# Continuing professional development opportunities for Australian endorsed for scheduled medicines podiatrists—What's out there and is it accessible, relevant, and meaningful? A cross-sectional survey

Saraid E. Martin[1,2]*, Kristin Graham[1], Helen A. Banwell[1], Jacinta L. Johnson[3,4]

**1** Allied Health & Human Performance, The University of South Australia, Adelaide, South Australia, Australia, **2** Podiatry Department, Southern Adelaide Local Health Network, Adelaide, South Australia, Australia, **3** Clinical & Health Sciences, The University of South Australia, Adelaide, South Australia, Australia, **4** SA Pharmacy, Statewide Clinical Support Services, SA Health, Adelaide, South Australia, Australia

* saraid.martin@mymail.unisa.edu.au

## Abstract

### Background

Non-medical prescribing is a valuable strategy to enable equitable access to medications in the context of the increasing demands on health services globally. Australian podiatrists have been able to seek endorsement for scheduled medicines (ESM) for over a decade. This project investigates the perceptions and habits of ESM podiatrists in meeting the extra continuing professional development (CPD) requirements associated with their ESM status.

### Methods

Australian ESM podiatrists completed an anonymous, online survey capturing demographics; CPD engagement; and self-reflections of CPD activities.

### Results

Twenty percent (n = 33) of Australian ESM registered podiatrists (N = 167) responded to the survey (18 female; median ESM status 2.5 years, (IQR 1.0, 9.0)). For the previous registration period, 88% (n = 29) completed the mandatory CPD hours, with only 35% (n = 11) completing a CPD learning goal plan. Over 80% identified their last ESM CPD activity as accessible, affordable, and could recommend to colleagues. Conversely, 50% or less agreed the activity increased confidence; changed their practice; improved communication skills; or enabled networking. Most respondents (81%, n = 27) indicated improvements should be made to the content, relevance, accessibility, and meaningfulness of CPD. These findings were supported by responses to the open-ended questions.

**Data Availability Statement:** All relevant data are within the paper and its Supporting Information files.

**Funding:** SM This work is supported by a Seed Funding Grant from the Allied and Scientific Health Office (ASHO), Department of Health and Wellbeing, SA Health. The funders had no role in study design, data collection and analysis, decision to publish, or preparation of the manuscript. https://www.sahealth.sa.gov.au/wps/wcm/connect/public+content/sa+health+internet/clinical+resources/clinical+governance+and+leadership/allied+and+scientific+health/research+and+evidence+informed+practice/allied+health+research+or+quality+improvement+project+seed+funding.

**Competing interests:** The authors have declared that no competing interests exist.

**Abbreviations:** ACPS, Australasian College of Podiatric Surgery; AHPRA, Australia Health Practitioner Regulation Agency; APodA, Australian Podiatry Association; CPD, Continuing professional development; ESM, Endorsed for scheduled medicines; NPS, National Prescribing Service; SA, South Australia; VIC, Victoria; QLD, Queensland; NSW, New South Wales; WA, Western Australia; ACT, Australian Capital Territory; NT, Norther Territory; TAS, Tasmania.

## Conclusions

Our findings suggest ESM podiatrists engage in CPD that is accessible rather than learning goal driven. Concerningly, CPD activities resulted in low translation of learnings to practice. This brings in to question the value of mandatory CPD systems based on minimum hours, rather than meaningfulness.

## Background

Non-medical prescribing is defined as the prescription of medicines by health care professionals who are not medical physicians or dentists [1]. Non-medical prescribing is a safe, clinically valuable, efficient, and cost-effective strategy for enabling health services to provide equitable and timely access to health care and medicines [2–7]. Furthermore, non-medical prescribing benefits health services, with improved staff satisfaction and increased retention of staff observed [1, 8].

First implemented in the United States of America in the 1960's [9] non-medical prescribing has since expanded across the globe with 99 countries identified as having a non-medical prescribing policy with varying scope of professions, practice, and restrictions in place [1–8, 10]. In Australia, non-medical prescribing enables nurses, midwives, and some allied health professions to prescribe medications from a discipline and state jurisdiction specific formulary once endorsement is obtained from the relevant legislative body [10]. To obtain endorsement for scheduled medicines (ESM), Australian podiatrists are required to successfully complete postgraduate qualifications [11]. Whilst it is an essential component of registration for podiatric surgeons, uptake of the ESM option remains low for non-surgeon podiatrists (167 of 5,867) [12]. However, as the Council of Australian Governments Health Council and the Podiatry Board of Australia have recently provided approval for an ESM training pathway to be embedded into undergraduate podiatry education nationally [13], ESM numbers are expected to rise.

The Podiatry Board of Australia requires those who have been successful in achieving ESM credentialling to undertake an additional 10 hours per year of continuing professional development (CPD) specific to endorsement (separate to the 20 hours of CPD required for standard podiatry registration) [14]. Anecdotally, ESM podiatrists have raised concerns over the perceived lack of availability and appropriateness of ESM CPD resources. While this may be a symptom of low ESM numbers in podiatry, access to appropriate, equitable, affordable, and relevant CPD opportunities are required to ensure safe, efficacious, and responsible prescribing practices continue as the profession evolves [15]. The expected increase in participation by podiatrists will bring with it the requirement to meet ongoing training and competency needs of this work force.

Worldwide, CPD is accepted as the minimum requirement for health care practitioners to update their knowledge, skills, and performance to maintain their clinical competency [16]. However, the true impact of CPD in enabling improved clinical service delivery is still largely unknown due to inconsistencies in assessing CPD outcomes [17, 18]. Mandatory CPD programs most commonly focus on hours completed and do not assess the content, nor quality of the activity undertaken. Several gaps in the CPD system are documented in the literature and range from an emphasis on a time-based approach over quality and a lack of focus of translation to practice of learnings, to disincentives to engage in multidisciplinary learning and team building activities [19, 20].

For CPD to be effective in building complex skills and changes to clinical practice that aim to ultimately improve patient care, CPD should be constructed based on adult learning

principles, rather than passive "tick a box" experiences [21]. Significant costs, both financial and time commitments, are associated with undertaking CPD, therefore it is imperative that the impact of CPD activities are adequately assessed [22, 23].

Kirkpatrick's Model, developed in the 1950's, is a widely accepted 4 stage model for evaluating training programs. Level 1 focuses on the learner's reaction to the training; level 2 evaluates content, skills, and knowledge; level 3 measures translation of knowledge and skills to practice; and level 4 measures impact and sustainability of the training [24]. Based on this model, evaluations of CPD often focus on a narrow range (levels 1 and 2) of easily measured impacts such as knowledge, skills, and confidence. Works by Allen et.al [17, 25, 26] generated the CPDIS tool that consists of three scales capturing "'learnings and self-efficacy', 'networking and building community', and 'achievement and validation'". Whilst these scales do encompass Kirkpatrick's four stages, they include impacts that are seldom measured and aim to capture a more wholistic evaluation of the effect of CPD [26].

With the continued implementation of non-medical prescribing models around the world and the expectation that many more podiatrists will obtain ESM status in the future, identifying factors associated with meeting CPD requirements, including whether podiatrists have adequate access to meaningful content are needed. Empirical evidence to support our understanding of these factors is essential to ensuring CPD experiences for podiatrists and non-medical prescribers alike, are both effective and appropriate to meet their needs.

The primary aim of this project was to investigate how ESM podiatrists meet the required CPD hours to maintain endorsement. The secondary aims were to determine if the learning opportunities currently available to ESM podiatrists are perceived by them to be accessible, relevant, and meaningful to achieve their desired learning goals.

## Methods

### Methodology

A descriptive, cross-sectional, anonymous online survey (Survey Monkey ™, California, USA) was conducted exploring ESM podiatrists' opinions and attitudes regarding CPD. Ethics approval was obtained from the Human Research Ethics Committee from the University of South Australia (Protocol number 203501). The study was performed in accordance with the ethical standards of the Declaration of Helsinki. All participants provided informed digital consent before commencing the survey.

### Participants

All Australian podiatrists with an endorsement for scheduled medicines and podiatric surgeons (N = 167 as of September 2021), were eligible for enrolment in this voluntary study [12]. Convenience sampling was utilised.

Potential participants were alerted to the study via several means. The research team utilised their extensive professional networks with invitations to participate and the survey URL link distributed via email and social media posts (Twitter™ and Facebook™). An advertisement was circulated to the Australasian College of Podiatric Surgery (ACPS) email distribution list. Participants were also encouraged to share the survey URL link amongst their ESM podiatry colleagues. No incentives were offered to potential participants and to prevent more than one entry from individuals, the multiple responses option within SurveyMonkey ™, which uses cookies, was turned off.

A detailed participant information sheet was displayed prior to the survey specifying the purpose of the study, approximate time to complete the survey, arrangements for the protection of personal information collected and data storage, and particulars of the investigative

team. Question one of the survey was mandatory and required all participants to provide informed digital consent before proceeding.

## Survey design

Data were collected using a purpose-built survey. The survey included a mixture of closed and open-ended questions relevant to the aims of the study. The final questionnaire consisted of 28 items over three sections (S1 File): Section 1: Focused on participant characteristics with data collected in line with the Podiatry Board of Australia's demographic reporting methods [12], with work sector, scope of practice and years of practice modified to be relevant to ESM prescribing. Data collected included gender, Podiatry Board registration type (ESM podiatrist, podiatric surgeon), Australian state or territory of most frequent practice, location of practice (metro or regional/rural/remote), primary work sector (private, public, community etc), work sector where they primarily prescribe, scope of practice (patient characteristics) they primarily prescribe for, years of practice and years as an ESM practitioner.

Section 2: Focused on participant CPD habits and engagement, with questions linked to engagement in ESM related CPD, mentoring related to ESM CPD, how participants planned for CPD and their preferences on the delivery of, and sectors responsible for ESM related CPD. Importantly, this section asked respondents to reflect on all the ESM CPD activities they had completed in the past 12 months and, on a 5-point Likert scale (where participants could indicate that they 'Strongly Agreed', 'Agreed', were 'Neutral', 'Disagreed' or 'Strongly Disagreed') rate their level of agreeance regarding ease of access, affordability, meaningfulness, relevance to scope of practice, improvement in knowledge, and translation to a change in prescribing practice. Questions were developed by the authorship group following a literature review of available relevant CPD surveys [20, 27–32] and discussion sessions with ESM podiatrists to ensure the construct was clearly understood. A final open-ended question was presented to participants asking them if they believed improvements could be made to the content, accessibility, relevance, and meaningfulness of CPD for ESM practitioners.

Section 3: Required participant self-reflection of the last ESM CPD activity they completed, including what the activity was, why they chose it, and if they would recommend the activity to others. As with section two, respondents were asked to evaluate the activity using a 5-point Likert scale with similar lines of enquiry, however the breadth of evaluation was expanded by utilising relevant items from the CPDIS tool [26] for question 25. Expanding the question base allowed participants to reflect on the broader impact of the CPD activity outside the easily and often measured Kirkpatrick's level 1 and 2 of skills and knowledge. As per the CPDIS tool, including measures such as networking and relationship building opportunities, change in practice, improved communication skills and shaping CPD and career development goals moves evaluation to Kirkpatrick's higher levels, and ensures deeper understanding of the quality of the CPD activity undertaken.

The final question was open-ended and provided participants the opportunity to comment on any other aspect of CPD for ESM podiatrists that they felt had not previously been covered.

The survey was pilot tested over two rounds by a convenience sample of five allied health clinicians (two podiatrists, an optometrist, pharmacist, and speech pathologist) who were asked to provide feedback regarding the structure, functionality, ambiguity, and face validity of the questions. The podiatrists involved in pilot testing were excluded from the final survey.

Questions were presented sequentially with participants able to navigate back within the browser to amend answers prior to completion. Key questions required mandatory answers to progress in the survey, and question logic was turned on to reduce the number of items posed to participants.

Survey results are reported following the CHERRIES (Checklist for Reporting Results of Internet E-Surveys) criteria [33] (S1 Table).

## Data collection

The online survey was open for participants to complete for five weeks between August and September 2021, at which point no further responses were recorded. No time stamp cut-off period was implemented for participants to complete the survey. To maximise completion rates, two email reminders were sent to invitees from the mailing list that had not responded to the survey by week 3 and again in the final week the survey was open.

## Data analysis

Data were exported from SurveyMonkey™ into a Microsoft Excel™ workbook. Descriptive data (age, years of registration and years of endorsement) were reviewed for normality. Responses to closed-ended questions were summarised using descriptive statistic calculations within Excel™. Having small numbers of responses, a pragmatic decision was made to condense the Likert scale responses into three groups: agree; neutral; and disagree [34–37].

Open-ended question responses underwent analysis using qualitative description. Similar to other qualitative methods, qualitative description is an 'empirical method of investigation aiming to describe the informant's perception and experience of the world and its phenomena' [38]. It differs from other qualitative methods as it uses a pragmatic approach to focus upon a straight description of an experience or an event. This method was selected as the open text responses received did not include sufficient depth to facilitate meaningful thematic analysis. Qualitative description analysis involved determination of a modifiable coding system that corresponded to the data collected. Analysis was conducted independently by two investigators with the coding system discussed until a consensus was reached. A low level of interpretation was applied to ensure the codes stayed close to the data, thus results are a description of informants' experiences in a language similar to the informants' own language [38, 39].

## Results

A total of 42 survey responses were received. Nine responses were excluded as participants were not ESM podiatrists or podiatric surgeons (n = 6) or failed to respond to questions outside of personal demographics (n = 3). The remaining 33 responses included in data analysis, represented 20% (n = 33/167) of the eligible population of ESM podiatrists and podiatric surgeons. Some survey participants skipped mandatory questions resulting in incomplete survey responses being received. Mandatory questions not involved in question logic that did not receive n = 33 responses were Q16 (n = 31); Q17 (n = 31); Q22–26 (n = 28).

Table 1 depicts the demographic data of participants. Respondents (n = 33) were largely ESM podiatrists (n = 26), the median years of practice was 11.5 years (IQR 6.8, 20.0), median years of endorsement was 2.5 (IQR 1.0, 9.0), and they were predominantly female (n = 18). Respondents were representative of all states and territories except the ACT and Tasmania.

## CPD habits and engagement of respondents

Participants had mixed views regarding the access to, and outcomes of ESM CPD. When asked how many hours of ESM CPD participants had completed in the previous 12-month registration period, 88% (n = 29) had undertaken the mandated 10 or more hours.

Participants perceptions regarding ease of access, affordability, meaningfulness, relevance to scope of practice of the CPD they had undertaken within the last 12 months, and its

**Table 1. Demographic details of respondents (n = 33).**

| Demographic characteristic | n (%) |
|---|---|
| Gender | |
| Male | 14 (42) |
| Female | 18 (55) |
| Prefer not to say | 1 (3) |
| Registration type | |
| General with endorsement for scheduled medicines | 26 (79) |
| Podiatric surgeon | 7 (21) |
| Jurisdiction of practice | |
| ACT | 0 |
| NSW | 4 (12) |
| NT | 1 (3) |
| QLD | 1 (3) |
| SA | 11 (33) |
| TAS | 0 |
| VIC | 12 (36) |
| WA | 4 (12) |
| Location of practice | |
| Metropolitan | 26 (79) |
| Regional rural or remote | 7 (21) |
| Primary work sector | |
| Private practice | 14 (42) |
| Community Health | 2 (6) |
| In-patient hospital | 6 (18) |
| Out-patient hospital | 6 (18) |
| Aged care | 0 |
| Disability services | 0 |
| Podiatric surgeon | 2 (6) |
| Other | 3 (9) |
| Primary work sector where prescribing occurs | |
| Private practice | 12 (36) |
| Community Health | 2 (6) |
| In-patient hospital | 0 |
| Out-patient hospital | 8 (24) |
| Aged care | 0 |
| Disability services | 0 |
| Podiatric surgeon | 5 (15) |
| Other | 6 (18) |
| Scope of practice medicines primarily prescribed | |
| High risk foot | 13 (39) |
| Paediatrics | 3 (9) |
| Sports | 1 (3) |
| General podiatry | 4 (12) |
| Aged care | 0 |
| Surgery | 11 (33) |
| Other | 1 (3) |
| Years of practice, median (IQR) | 11.5 (6.8, 20.0) |
| Years of endorsement, median (IQR) | 2.5 (1.0, 9.0) |

IQR = Interquartile range

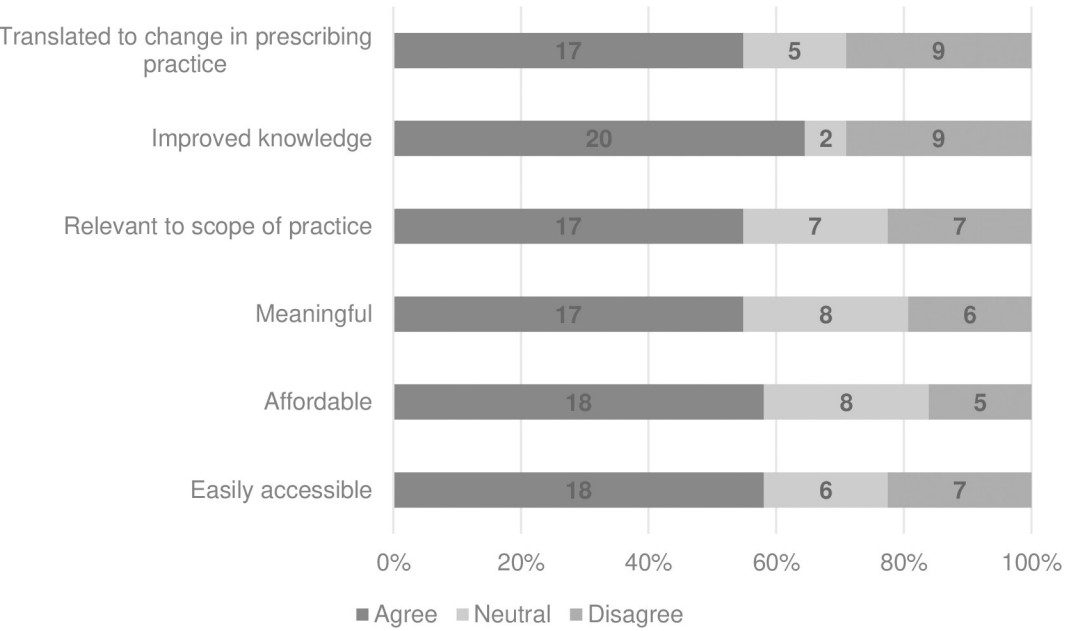

**Fig 1. Participant reflections of all endorsed for scheduled medicines continuing professional development activities completed in the past 12 months (n = 31).**

influence on knowledge and translation to a change in prescribing practice are summarised in Fig 1.

Overwhelmingly, 81% (n = 27) of respondents indicated there were improvements to be made to the content, accessibility, relevance, and meaningfulness of CPD available to ESM podiatrists.

Three further ideas arose from participants responses to the closed-ended questions regarding common behaviours and engagement strategies when undertaking CPD; a high level of mentoring exists within their cohort, most fail to set goals for CPD prospectively, and a wide variety of platforms are used to access CPD.

ESM podiatrists frequently seek and offer mentorship to peers. A majority 63% (n = 21) of respondents received mentoring from a more experienced prescriber. The main contributors to these mentoring relationships were ESM podiatrists (24%, n = 5), podiatric surgeons (29%, n = 6) and other medical specialists (33%, n = 7) such as infectious disease consultants or rheumatologists. Conversely, just over a third (36%, n = 12) of respondents had at some point mentored a podiatrist undertaking the ESM pathway to completion, with one participant reporting they had mentored 15 people through the education process. Ten respondents were mentoring a least one podiatrist undertaking the ESM pathway in the 2020 registration year, with one practitioner mentoring 10 pathway participants across the same period.

Nearly two-thirds (65%, n = 20) of respondents reported that they do not construct CPD learning goals at the start of a registration period, or they opt to write them reflectively after completing CPD activities.

Fig 2 outlines respondents' preferences for receiving CPD. The top three selected modalities were case studies, peer support sessions, and journal articles. Participants reported most often that they believe the Australian Podiatry Association (APodA) and multidisciplinary platforms should be delivering CPD activities for ESM clinicians (Fig 3).

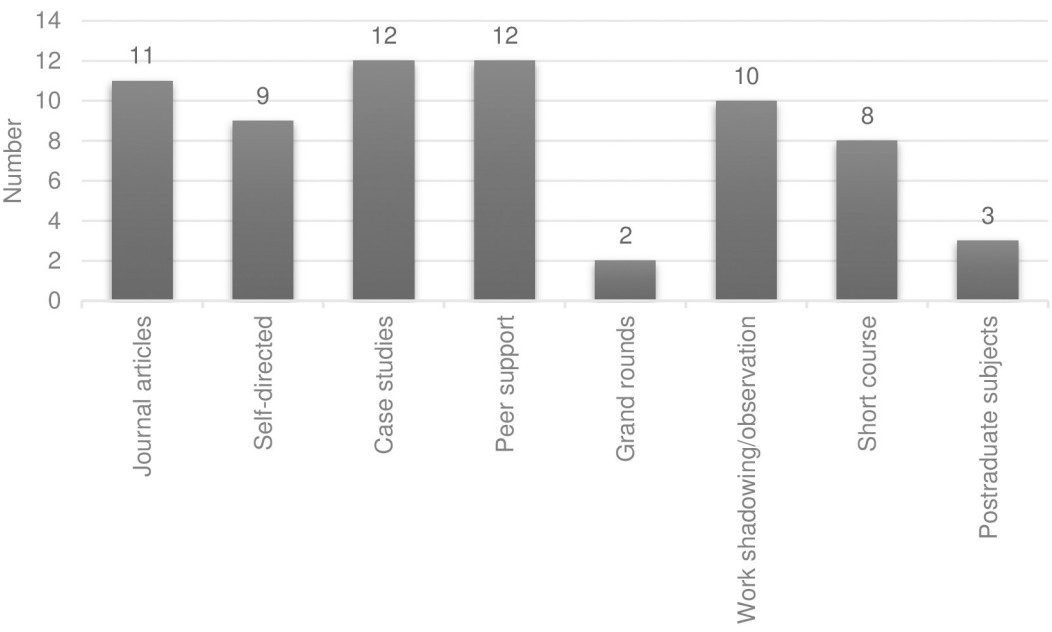

**Fig 2. Participant responses on how they prefer their continuing professional development to be delivered.** Participants (n = 33) were able to select their top 3 preferences.

## Self-reflection on participants last ESM CPD activity completed

Participants outlined the type of CPD activity they participated in, reasons for choosing the activity, and provided an assessment of the activity against accessibility, affordability, change in practice, and other skills.

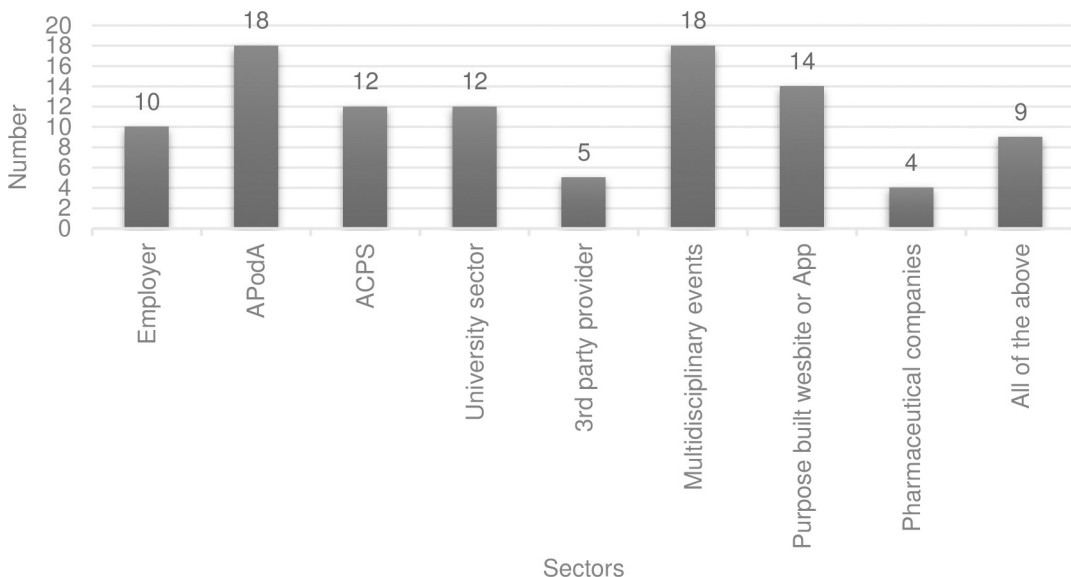

**Fig 3. Participant responses on which sectors they believe should be delivering continuing professional development for endorsed for scheduled medicines podiatrist.** Participants (n = 33) were able to select all that applied. (Note: APodA—Australia Podiatry Association. ACPS—Australasian College of Podiatric Surgery).

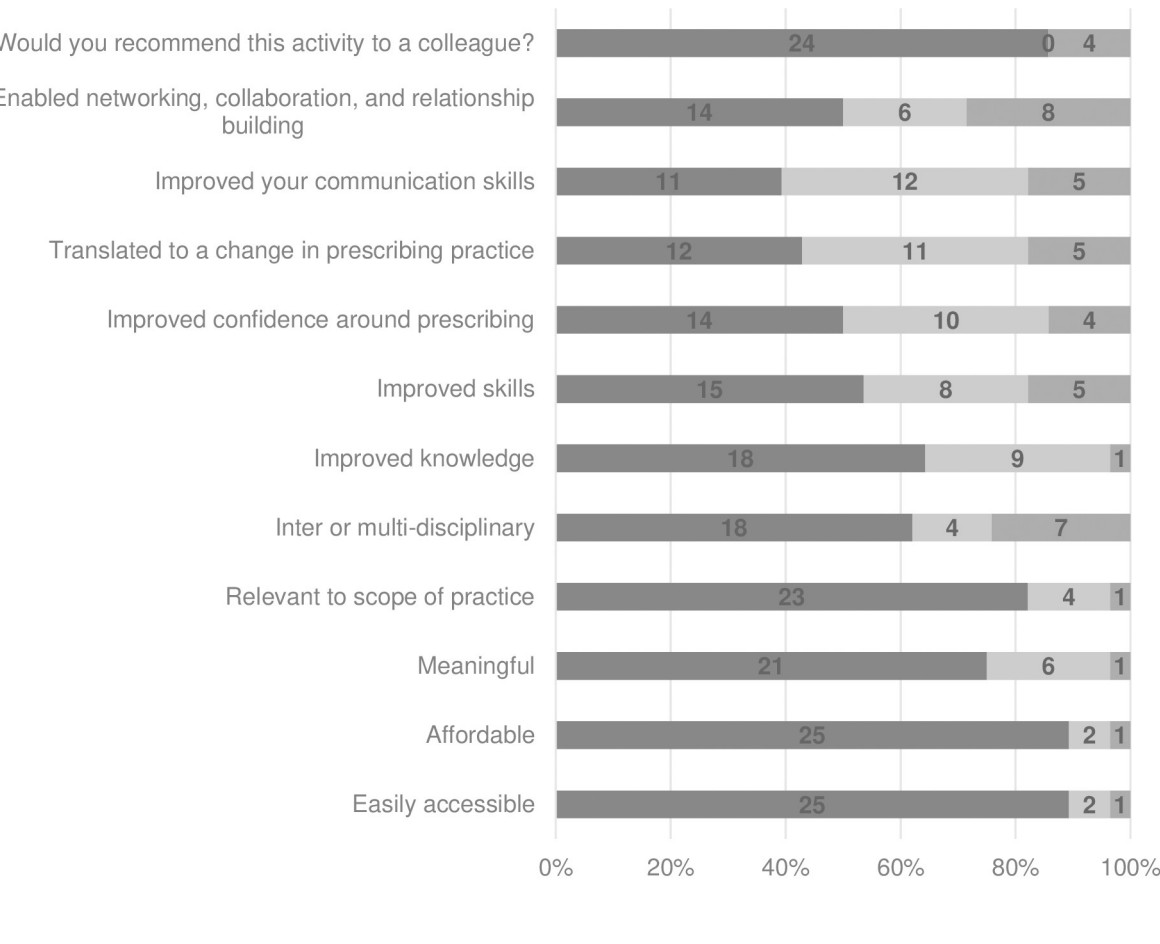

**Fig 4. Self-audit of the most recent endorsed for scheduled medicines continuing professional development activity undertaken (n = 28).** Participants were asked to rate on a 5-point Likert scale if the activity met the parameters outlined e.g. The activity enabled networking, collaboration and relationship building.

A range of CPD activities were reported by participants, including mentoring podiatrists undertaking the ESM pathway; reading COVID-19 updates, peer-reviewed journals, National Prescribing Service articles; accessing Therapeutic Guidelines; and taking part in or watching webinars.

The three most common reasons participants undertook their selected CPD activity was a clinical situation or interaction they had experienced (n = 13), it pertained to an area of interest (n = 13), or the ESM CPD activity was accessible (n = 13). No participants were prompted to undertake the activity due to a prescription audit or an error in prescribing.

Fig 4 outlines the respondent's self-assessment of the most recently completed ESM CPD activity against criteria adapted from the CPDIS tool [26]. The vast majority of respondents (89%, n = 25/28) reported that the activity was easily accessible and affordable and 85% (n = 24/33) would recommend the activity to a colleague. However, when asked if the activity improved confidence in their prescribing; translated to a change in practice; improved their communication skills with clients and/or colleagues; or enabled networking, collaboration and relationship building, only 50% or less of participants agreed.

### Descriptive analysis of open-ended questions

A total of seventy-nine responses (n = 79) were received for the two open-ended questions where participants were asked for their opinion on what improvements could be made to CPD for ESM practitioners. The *a priori* topic summaries of Accessibility, Meaningfulness, Content, and Relevance were well addressed by participants, with two new topic summaries identified as Delivery, and Networks and resources. Perhaps not surprisingly, the overwhelming message from respondents was;

"*We need more*!!" *(Podiatric surgeon, private practice, 12 years ESM).*

**Accessibility.** Access to appropriate CPD was repeatedly described as a barrier to ESM podiatrists. Of concern, finding forums to exchange information was highlighted:

"*Should be easier to give and obtain*" *(General podiatrist with ESM, hospital, 13 years practice, 9 years ESM).*

The time commitment required was a barrier to access:

"*I agree. . . there needs to be more CPD available that is easily accessible, meaningful. . . affordable and not too time consuming*" *(General podiatrist with ESM, hospital, 12 years practice, 7 years ESM).*

Similarly, one respondent found the workload required to maintain registration a barrier and lamented that this effort was not met by suitable rewards in the range of medications included on the prescribing formulary:

"*CPD requirements are far too onerous for the very limited prescribing schedule available*". *(Podiatric surgeon, private practice, 44 years practice, 10 years ESM).*

**Meaningfulness.** Respondents frequently commented on the lack of podiatry specific CPD that enabled meaningful learning experiences to occur:

"*There is VERY minimal out there relating to foot and ankle pathology and pharmacological management.*" *(General podiatrist with ESM, private practice, 13 years practice, 1 year ESM).*

Podiatry-led resources offered through the professional association were deemed by many respondents to be inadequate for their purpose:

"*Currently apoda (sic) cpd for this is repetitive*" *(General podiatrist with ESM, private practice, 8 years practice, 6 years ESM).*

Similarly, respondents perceived that the CPD activities on offer were trivial and cursory:

"*Diverse options needed. Most are passive at the moment and Tick the box, far more needed to be active learning*" *(General podiatrist with ESM, private practice, 27 years practice, 1 year ESM).*

As an alternative to podiatry led CPD, many participants reported utilising medical, nursing and pharmacy led resources which they often found irrelevant to podiatry practice:

*"At the moment . . .. information provided is directed at GP (sic) and often does not cover the medications that podiatrists can prescribe" (General podiatrist with ESM, private practice, 3 years practice, 1 year ESM).*

However, participants repeatedly described wanting CPD content that was meaningful to them:

*"Specifically aimed at endorsed podiatry prescribers." (General podiatrist with ESM, hospital, 19 years practice, 1 year ESM).*

**Content.** Generally, most respondents would like a broader range of content to be available. Some respondents had very specific areas of podiatry practice they wished to see ESM CPD content for, such as paediatrics, musculoskeletal, chronic disease, and high-risk foot. Additionally, many participants would value content addressing complex and high-risk scenarios:

*"High risk meds, special population prescribing, drug interactions" (General podiatrist with ESM, hospital, 8 years practice, 1 year ESM).*

The need for relevant, practical learning CPD activities was a frequent response. One participant highlighted that they valued CPD that took a broad case study approach enough to be prepared to pay for such CPD content:

*"It is hard to find CPD that is relevant to endorsed podiatry prescribers. I would happily pay for a distance type CPD approach that was just not re-iterating drug information, but was case study related to look at all aspects of the prescribing process" (General podiatrist with ESM, hospital, 19 years practice, 1 year ESM).*

**Delivery.** Flexible and variable delivery modes were preferred by participants. Online CPD resources and opportunities were commonly endorsed by respondents to improve access to CPD for ESM. This was especially relevant for those practitioners residing outside major metropolitan cities to ensure equity of access. Additionally, online delivery meets the need for flexible delivery styles, and the social distancing encouraged during the current COVID-19 related pandemic:

*"Zero events in remote areas, hence, online is the best alternative." (General podiatrist with ESM, private practice, 4 years practice, 2 years ESM).*

**Networks and resources.** Networks, relationships, and experiences was a topic summary that was identified from the data. This topic was highlighted as important to support clinicians to achieve their ESM CPD plans. There was acknowledgement from respondents that different prescribing settings and experiences could either facilitate or detract from their ability to engage with ESM CPD. A practitioners increased years of endorsement and connection with prescribing peers was thought to influence their ability to access resources more readily than isolated colleagues:

*"I think education is accessible to those that know where to find it, however, isolated podiatrists may not have awareness/confidence to access these resources." (Podiatric surgeon, private practice, 27 years practice, 10 years ESM).*

Differences in work sector was also seen as a contributing factor to accessing CPD resources successfully, with those in the public sector likely to be able to utilise multi and inter-disciplinary learning events compared to their colleagues working in private practice:

*"Many more opportunities for public podiatrists (access to grand rounds, peer shadowing) than private podiatrists." (General podiatrist with ESM, hospital, 4 years practice, 2 years ESM).*

Despite the critical appraisal of the current state of ESM CPD, participants provided many potential solutions. These included utilising existing resources such as mentors and podiatric surgeons as well as fostering inter-professional collaborations.

A focus on relationship building to strengthen support such as through extending the mentoring arrangements was suggested to assist practitioners to work to their full scope, particularly for those who are isolated:

"I suspect that a large sector of the general podiatry profession who gain ESM underutilise their new skills, due to isolated practice and/or policy limitations within their work environment. I feel that the mentoring arrangement should be continued beyond the time of graduation for ESM to ensure that each individual . . ..become effective prescribers" (Podiatric surgeon, private practice, 27 years practice, 10 years ESM).

Clinicians commented on specific existing resources that were deemed effective and working well, highlighted by the following statement:

*"The ACPS [Australasian College of Podiatric Surgeons] does provide pharmacology updates and education which enables uniformity in learning. Well organised and structured" (Podiatric surgeon, private practice, 34 years practice, 28 years ESM).*

This was further reinforced with many respondents articulating the need for podiatry specific CPD to be developed and delivered by podiatry peers as they will have profession specific content knowledge:

*"Frequent prescribers such as pod surgeons are better at delivering these courses as they do understand podiatry practice better." (Podiatric surgeon, private practice, 25 years practice, 13 years ESM).*

Finally, development of inter-professional networks to support and deliver CPD for ESM clinicians was also suggested with some respondents expressing a perceived benefit of linking into specialist professions such as pharmacy to deliver ESM CPD for podiatrists:

*"The area of pharmacy is so expansive that I feel it would be helpful if we had some support from the pharmacy profession to summarise all new drugs in the market and their potential interactions with our available medications" (Podiatric surgeon, private practice, 26 years practice, 9 years ESM).*

## Discussion

This survey presents the first known results describing how Australian ESM podiatrists are meeting the required CPD hours and if current learning opportunities are perceived to be

accessible, relevant, and meaningful. While respondents appreciated and sought out the knowledge associated with ESM activities, participants repeatedly called for improvements across these three domains. The overarching message was improvements across several parameters could be made, including frequency, content, relevance, accessibility, and delivery, with respondents suggesting solutions for many of the concerns raised. The results of the current study support previous international findings that CPD is perceived to be beneficial by clinicians [20, 28, 40–45]. There were also similarities shared between our study and those reported elsewhere for frustrations with scarcity of specialised content, cost and available funding, time commitment required, and the lack of national infrastructure to support quality CPD [8, 20, 27, 28, 32, 40–43].

The majority of respondents in our study reported that they were able to meet their mandatory CPD requirements by accessing a variety of education opportunities, across multiple platforms, with many of these experiences being viewed positively. However, only one-third of respondents reported that they complete CPD learning goals at the start of a registration period. Goal setting for CPD is considered best practice, recommended by Australian and international health bodies [19–21, 40, 42, 46–54] to ensure CPD is individualised, relevant to practice, and aids future career development [45]. An example of CPD goal setting being utilised on a large scale is the NHS annual professional development plans. However, outcomes in this case are hard to identify as participation rates in this process are not transparent [55–57]. Studies explicitly investigating clinician engagement with goal setting were not found. However, reflective practice, and, by extension, goal setting, is reportedly low across many health disciplines [45, 55, 58]. One explanation described in the literature is that clinicians are not trained in reflective practice or goal setting, particularly those who have been in the workforce longer [44].

Although podiatrists reported not explicitly setting goals for their CPD, they did report choosing activities based on 'clinical situations or interaction' or 'areas of interest'. This finding suggests an informal, implicit planning may be occurring when choosing CPD to engage with. While our survey did not capture the reasons for the lack of goal setting, several implications could be inferred; ESM podiatrists may not believe that planning results in a net benefit, there may be a lack of notification of appropriate CPD meaning they are unable to successfully plan their CPD for the coming year, or they fear that goal setting could impact their ability to achieve the mandatory CPD guideline requirements.

One reason reported by ESM podiatrists for completing CPD activities was ease of accessibility. Concerningly these accessible activities were often evaluated by participants as not providing relevant and meaningful content. International bodies governing or representing safe prescribing practice, advocate CPD should be undertaken with a focus on scope, and relevance to practice, aiming for the acquisition of knowledge and skills, and the ability to translate these into changes to the provision of clinical care [53, 54, 59]. Given CPD is mandatory, its effectiveness at achieving the required outcomes is worthy of examination, and to date the evidence in the broader literature of this effectiveness remains mixed [16]. In fact, while CPD has been shown to increase knowledge, translating this to practice change and safe delivery of quality care is yet to be clearly demonstrated [60].

Respondents to our survey provided numerous suggestions for improving ESM activities, including the creation of interprofessional networks. Interest in interprofessional learning has increased across the world with the concurrent shift to interprofessional and multidisciplinary patient centred health care models [61]. Interprofessional learning opportunities have demonstrated positive impacts that include improving interprofessional communication as well as an increase in earlier and appropriate referrals, and collaborative patient care delivery [61, 62]. Respondents suggested learning from and with pharmacists was a potentially valuable resource

that to date has not been fully explored. Fostering such interprofessional collaborative learning opportunities not only are likely to meet the learning needs of ESM podiatrists but may produce a change in behaviour by increasing the understanding of each other's professional roles, scopes, and improving communication [19].

A strong topic that emerged from this research was that some working environments provide broader opportunities for both structured and incidental CPD. For example, the isolated rural podiatrist may struggle to access adequate opportunities while those practicing in large multidisciplinary environments, surrounded by multiple prescribing practitioners may be at an advantage. It is well documented that rural health professionals experience barriers to engaging in professional development, including lengthy travel times to engage in events, a scarcity of backfill staffing to allow clinicians to attend events, and poor internet connections [63–65]. To ensure equity of access, CPD developers need to consider delivery modalities and the learning conditions of all potential participants regardless of their location.

With increasing demands on health services, the acceptance and support for non-medical prescribing is expected to rise [10]. Governments, professional advocacy, and regulatory bodies are encouraging clinicians internationally to become non-medical prescriber practitioners as a strategy to benefit consumers through improved access to medicines and to contribute to health care system efficiencies. In Australia, other allied health professions such as physiotherapy and pharmacy are currently exploring non-medical prescribing pathways for their professions. Preparing and planning for increasing numbers of endorsed non-medical prescribers is needed to ensure they can continue to practice in a safe, effective manner. This includes planning to support continuing professional development requirements associated with non-medical prescribing status.

Whilst these results provide some insight into ESM CPD experiences, there are limitations of this study that should be acknowledged. Firstly, only 20% of eligible ESM podiatrists responded to the survey. Only one response was received from a practitioner in Queensland, despite this jurisdiction having the second highest proportion (n = 42/167, 25%) of the overall ESM population [12]. Therefore, whilst these results provide some insight into ESM CPD experiences, they cannot be generalised to the entire ESM podiatry community. Secondly, on reflection, the survey design could have been strengthened by following high-quality survey development tools, such as those described by Artino et. al [66]. Thirdly, the authors also recognise that building questions around *a priori* topics may have limited the responses provided by participants. This was counteracted by the inclusion of a final open-ended question, "Do you have any other comments?" but may not have allowed participants to express all their views on CPD for ESM podiatrists. Further qualitative studies could add depth to the findings of this study.

From a medicine's perspective, the goal of CPD is to ensure clinicians maintain prescribing competency with the aim that this translates to safe, effective, and judicious use of medicines against the National Prescribing Service Prescribing Competencies Framework [66, 67]. It would be valuable for future research to examine the effectiveness of CPD activities for ESM podiatrists, and non-medical prescribers generally in achieving this goal. Shifting the focus of CPD evaluation more towards levels 3 and 4 of Kirkpatrick's Model and utilising tools such as CPDIS could assist in ascertaining if competency is achieved [17, 26]. Future work could also explore models for sustainable interprofessional learning and the effect of these programs on learning outcomes for each profession. With a projected increase in ESM practitioners in the future due to changes to undergraduate training programs and an increase in implementation of alternative models of health care, it will be important to explore changes in ESM practitioners learning requirements overtime. More broadly, utilisation of goal setting for CPD amongst

the podiatry population could be evaluated to ascertain if the results of this survey are consistent across the profession.

## Conclusions

The majority of ESM podiatrists do complete the 10 hours of mandatory training required by the Podiatry Board of Australia. However, some complete activities that are accessible and affordable, rather than being aligned to goals of learning, bringing into question the value of mandatory CPD. Respondents repeatedly called for improvements in CPD for ESM clinicians. They know what they want, how they want it delivered, and are prepared to invest in accessing relevant and quality resources to support their learning. ESM podiatrists would like podiatry led organisations and universities to deliver profession specific resources and develop interprofessional learning models that they can engage with. It is encouraging that these organisations have recognised the need to support ESM podiatrists in their CPD with new resources being recently released, however the challenge remains, as stated by clinicians in this survey "We want more!".

Whilst this study has focused on the Australian ESM podiatry context, learnings should prompt international jurisdictions that have non-medical prescribing in place to review the quality of ongoing educational support for practitioners. This is especially important for emerging prescribing professional groups or for countries planning to implement non-medical prescribing for the first time. Considering the continuing education needs of the non-medical prescribing workforce prior to implementation will assist in ensuring prescribers are supported robustly to provide safe, effective, and appropriate medicines care.

## Supporting information

**S1 File. Questionnaire.**
(PDF)

**S1 Table. CHERRIES checklist.**
(PDF)

**S2 Table. Questions 17, 24 &25 Likert scale analysis.**
(PDF)

**S1 Data.**
(XLSX)

## Acknowledgments

The authors wish to especially thank all endorsed for scheduled medicines podiatrists and podiatric surgeons who volunteered to take part in this study.

## Author Contributions

**Conceptualization:** Saraid E. Martin, Kristin Graham, Helen A. Banwell, Jacinta L. Johnson.

**Data curation:** Saraid E. Martin.

**Formal analysis:** Saraid E. Martin, Kristin Graham, Helen A. Banwell, Jacinta L. Johnson.

**Funding acquisition:** Saraid E. Martin.

**Investigation:** Saraid E. Martin.

**Methodology:** Saraid E. Martin, Kristin Graham, Helen A. Banwell, Jacinta L. Johnson.

**Project administration:** Saraid E. Martin.

**Resources:** Saraid E. Martin.

**Supervision:** Kristin Graham, Helen A. Banwell, Jacinta L. Johnson.

**Validation:** Saraid E. Martin, Helen A. Banwell, Jacinta L. Johnson.

**Writing – original draft:** Saraid E. Martin.

**Writing – review & editing:** Saraid E. Martin, Kristin Graham, Helen A. Banwell, Jacinta L. Johnson.

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
