## [Decision Letter · Decision Letter 0]

26 Apr 2023

PONE-D-23-05087

Continuing professional development opportunities for Australian endorsed for scheduled medicines podiatrists - what’s out there and is it accessible, relevant, and meaningful? A cross-sectional survey.

PLOS ONE

Dear Dr. Martin,

Thank you for submitting your manuscript to PLOS ONE. After careful consideration, we feel that it has merit but does not fully meet PLOS ONE’s publication criteria as it currently stands. Therefore, we invite you to submit a revised version of the manuscript that addresses the points raised during the review process.

Please address all comments raised by both reviewers.Please review the CHERRIES reporting format as identified by reviewer two and incorporate this into your data reporting.Of specific note, both reviewers identified issues with how the data is currently reported and the construction of the discussion. These points will require careful consideration in your review for the manuscript to be considered for publication.The discussion will need to be developed in a concise manner as both reviewers indicated the general nature and length of the discussion 

We look forward to receiving your revised manuscript.

Kind regards,

Matthew Carroll, *PhD., MEdL., MPod., BHSc, SFHEA*

Academic Editor

PLOS ONE

2. Please provide additional details regarding participant consent. In the ethics statement in the Methods and online submission information, please ensure that you have specified (1) whether consent was informed and (2) what type you obtained (for instance, written or verbal, and if verbal, how it was documented and witnessed). If your study included minors, state whether you obtained consent from parents or guardians. If the need for consent was waived by the ethics committee, please include this information. If you are reporting a retrospective study of medical records or archived samples, please ensure that you have discussed whether all data were fully anonymized before you accessed them and/or whether the IRB or ethics committee waived the requirement for informed consent. If patients provided informed written consent to have data from their medical records used in research, please include this information.

Reviewers' comments:

Reviewer's Responses to Questions

**Comments to the Author**

1. Is the manuscript technically sound, and do the data support the conclusions?

Reviewer #1: Yes

Reviewer #2: Partly

2. Has the statistical analysis been performed appropriately and rigorously? 

Reviewer #1: Yes

Reviewer #2: No

3. Have the authors made all data underlying the findings in their manuscript fully available?

Reviewer #1: Yes

Reviewer #2: Yes

4. Is the manuscript presented in an intelligible fashion and written in standard English?

Reviewer #1: Yes

Reviewer #2: Yes

5. Review Comments to the Author

Reviewer #1: #General comments

- I thank the authors for their efforts in conducting this research and preparing this manuscript.

- The authors present a cross-sectional survey of podiatry prescribers in Australia. Their aim is to understand how this population are utilising CPD, and their perceptions on accessibility, relevance, and meaningfulness. Appropriate ethics approval was obtained. I note that a pre-print has been published.

- The overall conceptual framework is clear and the methodology generally matches the research question. While I believe some aspects of the methodology could be improved, the final results appear to be novel and should be useful for informing CPD development for podiatry prescribers in Australia. However, as it is currently written, the findings are interpreted mainly for the Australian context, and more could be done to connect the results with broader literature and be more appealing to the international readership of PLOS ONE. I provide more detailed comments below:

#Introduction/background

- Clearly introduces the topic, the importance of non-medical prescribing, and the context of this in Australia.

- Unless I missed it, I do not believe ESM is written out in full next to the first time the abbreviation is used.

#Methods

- Study design is clear.

- Participants and recruitment is appropriate.

- The survey design process would have benefited from additional features of high-quality survey development. For example, refer to Artino et al 2014 AMEE Guide 87. Further, I would have liked to see additional use of theory to guide the development of the questions, although the authors refer to the CPDIS, after looking at their survey questions it is unclear to me how the CPDIS was used. This is important as linkage to theory would allow the authors to better discuss their findings relevant to the large body of CPD literature. I appreciate that it is now too late to go back to change the survey design process, however the authors could consider whether they could better describe some of the steps they did take with a view of convincing the reader of the quality of the survey tool, and also consider how this could be discussed in the limitations.

- Data analysis, the analysis methods for the qualitative data is not clear to me. I would have liked reference to a specific qualitative analysis approach so that it can be understood how the analysis actually took place.

#Results

- The results text reports career length using mean, but Table 1 reports as median, consider if this should be made consistent.

- Is it possible to report the n of respondents for each state together with the total n of ESM in the state? I suspect this data should be available and would provide context for interpreting the numbers.

- I am concerned that some participants undertook <10 hrs of CPD given in the introduction you’ve stated that at least 10 hrs is a requirement for registration.

- Consider if it would be more useful to provide all 5-pts of the 5-pt Likert scale in your Figures. I believe you would have space. I would also find it helpful is the exact n of responses for each category was also included on the figure. This can be easily overlayed onto the bar chart.

- In the results you use the word ‘theme’ several times. To me, this implies the use of thematic analysis, which I do not believe you have done. I suggest using another word if you did not actually use a theme-generating method.

#Discussion/Conclusion

- There is an absence of reference to other literature in the discussion. I would like to see the results described relative to other studies looking at essentially ‘satisfaction’ with CPD. This could be in other countries, other professions, or other topics. I would like to know whether the findings are unique to pod ESM CPD, or whether this is a common problem of CPD in general.

- There is a very strong Australian podiatry focus in the discussion which would be better suited to a podiatry specific journal. I’d like to see greater consideration of how providing CPD is considered by other professions and in other jurisdictions to engage a wider readership. The underlying concept appears to be CPD access in extended scope areas.

- In one paragraph you refer to a “dearth of access to mentors”. This does not appear consistent with the results showing strong use of mentors, and I do not believe you capture data that would allow you to make any claim about access to mentors.

#Writing

- Overall a clear writing style. There is a fair bit of content in the manuscript owing the range of ideas looked at in the survey, which did make it tricky to follow at times. I believe with some careful consideration of what content is essential to the overall message from the manuscript, as well as editing, the word count for the existing content could be reduced by 15-20%, particularly in the discussion. This would allow for inclusion of some of my suggestions without making the manuscript too long.

Reviewer #2: Introduction

Relevant concepts er discussed to develop the rationale. One concept introduced in the last section of the discussion but not developed in the introduction is the relevance of the Kirkpatrick model. I have made further comments on this in the discussion section

Data collection

Can you please refer to the Checklist for Reporting Results of Internet E-Surveys (CHERRIES). Eysenbach G. Improving the quality of Web surveys: the Checklist for Reporting Results of Internet E-Surveys (CHERRIES). J Med Internet Res. 2004 Sep 29;6(3):e34. doi: 10.2196/jmir.6.3.e34. Erratum in: doi:10.2196/jmir.2042. PMID: 15471760; PMCID: PMC1550605.

Please complete the CHERRIES form and include it as a supplementary file

There are statements that require clarity in this section that will be aided by the completion and reporting of your results in alignment with CHERRIES. For example, you state “ Email reminders were sent to targeted participants to maximise completion rates” you will need to clarify how many and to what groups they were sent. You also note there was no time stamp cut-off period was implemented for participants to complete the survey. I assume that each participant had 5 weeks to complete the survey from the date they opened it? As opposed to the survey closing for all participants after 5 weeks.

Data analysis

Please report all Likert-based data in a Table as median with IQR. You have compacted the results into three-item stacked bar charts but this provides no representation of the mean or spread of data. This comment applies to (Q17, 24, 25). Could you also consider a between-group analysis (ESM podiatrists vs Podiatric surgeons) as this would add some richness to the data?

What methods were used to analyse open-ended responses

Results

Overall the results lack clarity and consistency of reporting. Please review all data for consistency and appropriateness with particular attention to measures of central tendency (mean vs median)

You will also need to add some of the detail to clarify the reporting of the survey results into the results section. Currently, it is very unclear surrounding detail for example incomplete responses vs non-responses. Completion of the CHERRIES form will aid this process.

In Figures 1 and 4 you present data based on agree, neutral or disagree yet your questions used a 5-point Likert scale. Your methods provide no information surrounding the handling of this data.

You state respondents were representative of all states and territories except the ACT and Tasmania. This should be reworded to responses were received from all states.

Discussion

Much of the discussion is a repeat of the results. Consequently, it is unclear what the most significant and important findings are. The discussion tries to encompass all results rather than narrowing down to the most important or interesting findings. Following numerous read-throughs it is still unclear what the most important results were. The discussion could be written more concisely around three or four major findings as currently, the discussion is very lengthy.

More consideration of other work/research that has been published around CPD and podiatry is required. There is no discussion or comparison to other countries.

The discussion refers to the Kirkpatrick model specifically shifting CPD evaluation towards levels 3 and 4. This paragraph lacks underpinning, the readers will have no knowledge unless you introduce the model and what levels 3 and 4 would represent in terms of the ESM CPD framework. Consider how you will introduce the readers to the Kirkpatrick model, you may need to do this in the introduction.

6. PLOS authors have the option to publish the peer review history of their article (what does this mean?). If published, this will include your full peer review and any attached files.

Reviewer #1: **Yes: **Jonathan Foo

Reviewer #2: No

---

## [Author Response · Author response to Decision Letter 0]

17 Jun 2023

All responses to comments are included in the attached Response to Reviewers document. 

I wish to extend my thanks to the Academic Editor and both Peer Reviewers for taking the time to assess our article and provide constructive suggestions for improvement.

JOURNAL REQUIREMENTS:

Please ensure that your manuscript meets PLOS ONE's style requirements, including those for file naming

 The manuscript has been edited to meet PLOS ONE’s style requirements.

Please provide additional details regarding participant consent. In the ethics statement in the Methods and online submission information, please ensure that you have specified:

(1) whether consent was informed and 

(2) what type you obtained (for instance, written or verbal, and if verbal, how it was documented and witnessed). 

If your study included minors, state whether you obtained consent from parents or guardians. 

If the need for consent was waived by the ethics committee, please include this information. 

If you are reporting a retrospective study of medical records or archived samples, please ensure that you have discussed whether all data were fully anonymized before you accessed them and/or whether the IRB or ethics committee waived the requirement for informed consent. 

If patients provided informed written consent to have data from their medical records used in research, please include this information.

 Methods section, line 132 states:

All participants provided informed digital consent before commencing the survey.

Methods section, line 147 states:

Question one of the survey was mandatory and required all participants to provide informed digital consent before proceeding.

Minors were not eligible to participate in this study. Medical records were not accessed as part of this study.

The above details have also been added to the online submission information. 

Your ethics statement should only appear in the Methods section of your manuscript. If your ethics statement is written in any section besides the Methods, please move it to the Methods section and delete it from any other section. Please ensure that your ethics statement is included in your manuscript, as the ethics statement entered into the online submission form will not be published alongside your manuscript.

 Methods section, Line 129 of the manuscript states:

Ethics approval was obtained from the Human Research Ethics Committee from the University of South Australia (Protocol number 203501). The study was performed in accordance with the ethical standards of the Declaration of Helsinki.

We have deleted the Ethics approval and consent to participate statement from the Declarations section. 

REVIEWER 1:

Unless I missed it, I do not believe ESM is written out in full next to the first time the abbreviation is used.

 Thank you for pointing out the abbreviation error. This has now been corrected in Background, line 77.

The survey design process would have benefited from additional features of high-quality survey development. For example, refer to Artino et al 2014 AMEE Guide 87. Further, I would have liked to see additional use of theory to guide the development of the questions, although the authors refer to the CPDIS, after looking at their survey questions it is unclear to me how the CPDIS was used. This is important as linkage to theory would allow the authors to better discuss their findings relevant to the large body of CPD literature. I appreciate that it is now too late to go back to change the survey design process, however the authors could consider whether they could better describe some of the steps they did take with a view of convincing the reader of the quality of the survey tool, and also consider how this could be discussed in the limitations.

 We have expanded the explanation for how the survey items were developed with the following additions to the manuscript:

Line 168 –

Questions were developed by the authorship group following a literature review of available relevant CPD surveys (20, 27, 28, 29, 30, 31, 32) and discussion sessions with ESM podiatrists to ensure the construct was clearly understood. 

Line 175 – 

As with section two, respondents were asked to evaluate the activity using a 5-point Likert scale with similar lines of enquiry, however the breadth of evaluation was expanded by utilising relevant items from the CPDIS tool (26) for question 25. Expanding the question base allowed participants to reflect on the broader impact of the CPD activity outside the easily and often measured Kirkpatrick’s level 1 and 2 of skills and knowledge. As per the CPDIS tool, including measures such as networking and relationship building opportunities, change in practice, improved communication skills and shaping CPD and career development goals moves evaluation to Kirkpatrick’s higher levels, and ensures deeper understanding of the quality of the CPD activity undertaken.

The design of the survey has also been discussed as a limitation:

Discussion, line 494 –

On reflection, the survey design could have been strengthened by following high-quality survey development tools, such as those described by Artino et. al (2014). The authors also recognise that building questions around a priori topics may have limited the responses provided by participants. 

Data analysis, the analysis methods for the qualitative data is not clear to me. I would have liked reference to a specific qualitative analysis approach so that it can be understood how the analysis actually took place.

 Description of the methodology used has been strengthened with the following added to the manuscript:

Line 207:

Open-ended question responses underwent analysis using qualitative description. Similar to other qualitative methods, qualitative description is an ‘empirical method of investigation aiming to describe the informant's perception and experience of the world and its phenomena’ (38). It differs from other qualitative methods as it uses a pragmatic approach to focus upon a straight description of an experience or an event. This method was selected as the open text responses received did not include sufficient depth to facilitate meaningful thematic analysis. Qualitative description analysis involved determination of a modifiable coding system that corresponded to the data collected. Analysis was conducted independently by two investigators with the coding system discussed until a consensus was reached. A low level of interpretation was applied to ensure the codes stayed close to the data, thus results are a description of informants' experiences in a language similar to the informants' own language (38, 39). 

The results text reports career length using mean, but Table 1 reports as median, consider if this should be made consistent.

 The text has been updated to reflect results in Table 1. 

Line 226:

Table 1 depicts the demographic data of participants. Respondents (n = 33) were largely ESM podiatrists (n = 26), their median years of practice was 11.5 years (IQR 6.8, 20.0), median years of endorsement was 2.5 (IQR 1.0, 9.0), and they were predominantly female (n = 18).

Is it possible to report the n of respondents for each state together with the total n of ESM in the state? I suspect this data should be available and would provide context for interpreting the numbers.

 This data is indeed available, and we had considered reporting against the total n of ESM in the state. However, as total n is low for some states, this could, in-turn, identify the survey participant. 

I am concerned that some participants undertook <10 hrs of CPD given in the introduction you’ve stated that at least 10 hrs is a requirement for registration.

 Yes, this was a concerning result, but one we felt important to report. This finding demonstrates the importance of this work to ensure that CPD available to ESM podiatrists is engaging to encourage and assist them to meet the minimum registration requirements. 

All data was anonymous and therefore participants are not identifiable, and as a result of this publication will not be subject to further investigation into their registration declarations. 

Consider if it would be more useful to provide all 5-pts of the 5-pt Likert scale in your Figures. I believe you would have space. I would also find it helpful is the exact n of responses for each category was also included on the figure. This can be easily overlayed onto the bar chart.

 Thank you for prompting further thought and discussion on our reporting of the Likert scale responses. The research team did consider your suggestion of reporting all 5-points of the Likert scale, however due to small numbers we made the pragmatic decision to pool the responses. This is not uncommon in the literature, and I have included in the manuscript an explanation and supporting literature sources to justify the decision.

Line 204: 

Having small numbers of responses, a pragmatic decision was made to condense the Likert scale responses into three groups: agree; neutral; and disagree 34, 35, 36, 37).

The number of responses for the three combined groups have been included in figures 1 & 4. 

In the results you use the word ‘theme’ several times. To me, this implies the use of thematic analysis, which I do not believe you have done. I suggest using another word if you did not actually use a theme-generating method.

 As you correctly state thematic analysis has not been used to analyse the data. Thank you for highlighting that using the word ‘theme’ when not engaging with TA can lead to confusion. As such, we have changed the terminology throughout the results section to topic summaries as suggested by Braun & Clarke (2022).

Line 302:

The a priori topic summaries of Accessibility, Meaningfulness, Content, and Relevance were well addressed by participants, with two new topic summaries identified as Delivery, and Networks and resources.

There is an absence of reference to other literature in the discussion. I would like to see the results described relative to other studies looking at essentially ‘satisfaction’ with CPD. This could be in other countries, other professions, or other topics. I would like to know whether the findings are unique to pod ESM CPD, or whether this is a common problem of CPD in general.

 We have included international literature of other medicines prescribers and their views regarding CPD to support their practice. 

Discussion, Line 421:

Like ESM podiatrists, internationally CPD is perceived to be beneficial by clinicians, however this does not always correlate with satisfaction of all or some aspects of the CPD on offer. Many studies reported frustration amongst prescribers regarding the scarcity of specialised content, cost and available funding, time commitment required to complete CPD and the lack of national infrastructure to support quality CPD (8, 20, 27, 28, 32, 40, 41, 42, 43.

In one paragraph you refer to a “dearth of access to mentors”. This does not appear consistent with the results showing strong use of mentors, and I do not believe you capture data that would allow you to make any claim about access to mentors.

 To condense the discussion section, we have deleted this statement. However, access to mentors is indeed restricted for ESM podiatrists and other non-medical prescribers internationally. This has been widely reported as a barrier to undertaking the endorsement pathway/qualification. The data collected in this study highlights that whilst many ESM podiatrists engage in ongoing mentorship post-endorsement, often it is the same few people who provide that mentorship. 

Overall a clear writing style. There is a fair bit of content in the manuscript owing the range of ideas looked at in the survey, which did make it tricky to follow at times. I believe with some careful consideration of what content is essential to the overall message from the manuscript, as well as editing, the word count for the existing content could be reduced by 15-20%, particularly in the discussion. This would allow for inclusion of some of my suggestions without making the manuscript too long.

 Thank you for the feedback. In particular, we have revised the discussion section to be more succinct in its reporting and to incorporate international perspectives and literature. 

REVIEWER 2:

Relevant concepts are discussed to develop the rationale. One concept introduced in the last section of the discussion but not developed in the introduction is the relevance of the Kirkpatrick model. I have made further comments on this in the discussion section

 Thank you for pointing this out. We have introduced the Kirkpatrick Model in the introduction and the concept of adult learning principles as suggested. 

Line 105:

Kirkpatrick’s Model, developed in the 1950’s, is a widely accepted 4 stage model for evaluating training programs. Level 1 focuses on the learner’s reaction to the training; level 2 evaluates content, skills, and knowledge; level 3 measures translation of knowledge and skills to practice; and level 4 measures impact and sustainability of the training (24).

Please complete the CHERRIES form and include it as a supplementary file

 This has been completed and is attached as supplementary file 2. 

There are statements that require clarity in this section that will be aided by the completion and reporting of your results in alignment with CHERRIES. For example, you state “ Email reminders were sent to targeted participants to maximise completion rates” you will need to clarify how many and to what groups they were sent. You also note there was no time stamp cut-off period was implemented for participants to complete the survey. I assume that each participant had 5 weeks to complete the survey from the date they opened it? As opposed to the survey closing for all participants after 5 weeks.

 We have added additional information to these statements:

Methods, line 196: 

The online survey was open for participants to complete for five weeks between August and September 2021, at which point no further responses were recorded No time stamp cut-off period was implemented for participants to complete the survey. To maximise completion rates, two email reminders were sent to invitees from the mailing list that had not responded to the survey at week at 3 and in the final week the survey was open.

Please report all Likert-based data in a Table as median with IQR. You have compacted the results into three-item stacked bar charts but this provides no representation of the mean or spread of data. This comment applies to (Q17, 24, 25). 

 We have opted to leave the bar charts in the manuscript as we feel that the data is more digestible for clinicians and CPD providers who are not necessarily research trained. However, supporting file 3 has been added, which contains the Likert-based data (Q 17, 24, 25) in a table with median and IQR described. 

Could you also consider a between-group analysis (ESM podiatrists vs Podiatric surgeons) as this would add some richness to the data?

 Whilst we agree, including a sub-group comparison between ESM podiatrists and podiatric surgeons could be interesting, the sample size for podiatric surgeons is small (n=7) and is in a non-normally distributed data set. This would make drawing a meaningful conclusion difficult, and we have therefore opted not to include this data. 

What methods were used to analyse open-ended responses

 As replied above, Qualitative description was used. Further details have been provided regarding this choice.

Line 207:

Open-ended question responses underwent analysis using qualitative description. Similar to other qualitative methods, qualitative description is an ‘empirical method of investigation aiming to describe the informant's perception and experience of the world and its phenomena’ (38). It differs from other qualitative methods as it uses a pragmatic approach to focus upon a straight description of an experience or an event. This method was selected as the open text responses received did not include sufficient depth to facilitate meaningful thematic analysis. Qualitative description analysis involved determination of a modifiable coding system that corresponded to the data collected. Analysis was conducted independently by two investigators with the coding system discussed until a consensus was reached. A low level of interpretation was applied to ensure the codes stayed close to the data, thus results are a description of informants' experiences in a language similar to the informants' own language (38, 39). 

Overall the results lack clarity and consistency of reporting. Please review all data for consistency and appropriateness with particular attention to measures of central tendency (mean vs median)

 As replied above:

Line 226:

Table 1 depicts the demographic data of participants. Respondents (n = 33) were largely ESM podiatrists (n = 26), their median years of practice was 11.5 years (IQR 6.8, 20.0), median years of endorsement was 2.5 (IQR 1.0, 9.0), and they were predominantly female (n = 18).

You will also need to add some of the detail to clarify the reporting of the survey results into the results section. Currently, it is very unclear surrounding detail for example incomplete responses vs non-responses. Completion of the CHERRIES form will aid this process.

 See method section responses above and supporting file 2. 

Line 222:

Some survey participants skipped mandatory questions resulting in incomplete survey responses being received. Mandatory questions not involved in question logic that did not receive n=33 responses were Q16 (n=31); Q17 (n=31); Q22 - 26 (n=28).

In Figures 1 and 4 you present data based on agree, neutral or disagree yet your questions used a 5-point Likert scale. Your methods provide no information surrounding the handling of this data.

 As replied above:

Line 204: 

Having small numbers of responses, a pragmatic decision was made to condense the Likert scale responses into three groups: agree; neutral; and disagree 34, 35, 36, 37).

You state respondents were representative of all states and territories except the ACT and Tasmania. This should be reworded to responses were received from all states.

 Responses were received from all states and territories, except the ACT and Tasmania. One participant identified as practicing in the Northern Territory which is an Australian territory, therefore rewording to “received from all states” would be inaccurate. 

Much of the discussion is a repeat of the results. Consequently, it is unclear what the most significant and important findings are. The discussion tries to encompass all results rather than narrowing down to the most important or interesting findings. Following numerous read-throughs it is still unclear what the most important results were. The discussion could be written more concisely around three or four major findings as currently, the discussion is very lengthy.

 Thank you for these constructive comments.

The discussion has been re-written, with the 4 main findings as follows:

1. CPD is being completed but not in a structed way

2. CPD activities are often chosen as they are accessible, not because they are relevant or meaningful

3. Suggestions for improvements to CPD

4. Different learning environments can be a barrier.

More consideration of other work/research that has been published around CPD and podiatry is required. There is no discussion or comparison to other countries.

 International perspectives and literature has now been included in this section. See responses above for further details.

Discussion, Line 421:

Like ESM podiatrists, internationally CPD is perceived to be beneficial by clinicians, however this does not always correlate with satisfaction of all or some aspects of the CPD on offer. Many studies reported frustration amongst prescribers regarding the scarcity of specialised content, cost and available funding, time commitment required to complete CPD and the lack of national infrastructure to support quality CPD (8, 20, 27, 28, 32, 40, 41, 42, 43.

The discussion refers to the Kirkpatrick model specifically shifting CPD evaluation towards levels 3 and 4. This paragraph lacks underpinning, the readers will have no knowledge unless you introduce the model and what levels 3 and 4 would represent in terms of the ESM CPD framework. Consider how you will introduce the readers to the Kirkpatrick model, you may need to do this in the introduction.

 This discussion has been moved to the introduction as suggested. 

Line 105.

Kirkpatrick’s Model, developed in the 1950’s, is a widely accepted 4 stage model for evaluating training programs. Level 1 focuses on the learner’s reaction to the training; level 2 evaluates content, skills, and knowledge; level 3 measures translation of knowledge and skills to practice; and level 4 measures impact and sustainability of the training (24).

Please upload your figure files to the Preflight Analysis and Conversion Engine (PACE) digital diagnostic tool

 Figure files have been added to the PACE tool as requested.

---

## [Decision Letter · Decision Letter 1]

14 Jul 2023

Continuing professional development opportunities for Australian endorsed for scheduled medicines podiatrists - what’s out there and is it accessible, relevant, and meaningful? A cross-sectional survey.

PONE-D-23-05087R1

Dear Dr. Martin,

We’re pleased to inform you that your manuscript has been judged scientifically suitable for publication and will be formally accepted for publication once it meets all outstanding technical requirements.

Kind regards,

Matthew Carroll, PhD., MEdL., MPod., BHSc., SFHEA

Academic Editor

PLOS ONE

Additional Editor Comments (optional):

Please note reviewer 1 has made some comments for consideration of the authorship team. However you are not required to addresss any of these comments.

Reviewers' comments:

Reviewer's Responses to Questions

**Comments to the Author**

1. If the authors have adequately addressed your comments raised in a previous round of review and you feel that this manuscript is now acceptable for publication, you may indicate that here to bypass the “Comments to the Author” section, enter your conflict of interest statement in the “Confidential to Editor” section, and submit your "Accept" recommendation.

Reviewer #1: (No Response)

Reviewer #2: All comments have been addressed

2. Is the manuscript technically sound, and do the data support the conclusions?

Reviewer #1: Yes

Reviewer #2: (No Response)

3. Has the statistical analysis been performed appropriately and rigorously? 

Reviewer #1: Yes

Reviewer #2: (No Response)

4. Have the authors made all data underlying the findings in their manuscript fully available?

Reviewer #1: Yes

Reviewer #2: (No Response)

5. Is the manuscript presented in an intelligible fashion and written in standard English?

Reviewer #1: Yes

Reviewer #2: (No Response)

6. Review Comments to the Author

Reviewer #1: Thank you for the revisions made. I can see that you’ve taken on board the suggestions and made significant improvements to this manuscript. In particular, the methods section is now clear.

I would like to encourage you to again consider refining your writing and how you craft the narrative of your manuscript. As previously mentioned, there are a lot of ideas presented which makes it difficult to follow. While it is much improved, I believe it can be improved further. Firstly, at a writing craft level, this includes considering how you structure your sentences to aid alignment between sections and logical flow within sections. There are a number of peripheral sentences that could be removed or combined without affecting the overall ideas of the study, and only serve to distract the reader. Secondly, at a conceptual level, this can be done through thinking critically about other literature and your results, in particular considering how to present your findings such that they are most useful to others who want to continue improving CPD. I provide a few specific examples below:

Line 433 – as a clinician, I am not surprised at the lack of goal setting and suggest that the rate you reported may be very similar to general health professionals. While you suggest a number of reasons why this may be happening, it would be helpful for you to link your suggestions to literature, considering whether anyone else has specifically researched the topic of lack of goal setting in CPD.

Line 438 – your claim here that CPD activity is selected based on accessibility does not appear to match the information reported in the results. “The three most common reasons participants undertook their selected CPD activity was a clinical situation or interaction they had experienced (n = 13), it pertained to an area of interest (n = 13), or the ESM CPD activity was accessible (n = 13).” To me, this makes accessibility equal to a particular learning need (i.e. experience) and interest, but in your discussion you’ve neglected to mention these reasons. Please consider if you’ve overinterpreted results to fit your desired narrative, and whether this needs to be moderated somewhat. I also note that this same interpretation is made in the conclusion Line 516.

Line 445 – you state the evidence for CPD is mixed, but I would like to see a more critical appraisal of the literature related to your study. For example, Line 440 mentions that CPD was found to be not relevant or meaningful. How does this match with previous literature? Is the issue of relevance or meaningfulness unique to ESM CPD? Or is it an issue more broadly in CPD? This is critical as it informs how we think about improving the situation.

Line 490 – to better cater for the international readership, consider whether it is useful to specifically state that no responses were received from TAS and ACT. The state boundaries will mean little to anyone outside of Australia, and even as somebody in Australia I cannot make any real meaning from these sentences (i.e. what is the implication?) My suggestion is to either a) remove specific mention of states in the discussion and simply refer to the limitation that responses were not received from all jurisdictions, or b) make some effort to explain why it may be the case that no responses were received. Additionally, for the limitations paragraph, the writing would be easier to read if you signpost for each separate limitation. E.g. Firstly… Secondly… Thirdly…

Once again, congratulations on your work. I am satisfied with the revisions and am happy to defer to the editors decision regarding the final manuscript. I do not need to review this again.

Reviewer #2: (No Response)

7. PLOS authors have the option to publish the peer review history of their article (what does this mean?). If published, this will include your full peer review and any attached files.

Reviewer #1: **Yes: **Jonathan Foo

Reviewer #2: **Yes: **Matthew R. Carroll

---

## [Editor Report · Acceptance letter]

11 Sep 2023

PONE-D-23-05087R1 

Continuing professional development opportunities for Australian endorsed for scheduled medicines podiatrists - what’s out there and is it accessible, relevant, and meaningful? A cross-sectional survey. 

Dear Dr. Martin:

I'm pleased to inform you that your manuscript has been deemed suitable for publication in PLOS ONE. Congratulations! Your manuscript is now with our production department. 

Kind regards, 

on behalf of

Associate Professor Matthew Carroll 

Academic Editor

PLOS ONE